# Novel Hydrogen Sulfide Hybrid Derivatives of Keap1-Nrf2 Protein–Protein Interaction Inhibitor Alleviate Inflammation and Oxidative Stress in Acute Experimental Colitis

**DOI:** 10.3390/antiox12051062

**Published:** 2023-05-08

**Authors:** Xian Zhang, Keni Cui, Xiaolu Wang, Yuanyuan Tong, Chihong Liu, Yuechao Zhu, Qidong You, Zhengyu Jiang, Xiaoke Guo

**Affiliations:** 1State Key Laboratory of Natural Medicines and Jiang Su Key Laboratory of Drug Design and Optimization, China Pharmaceutical University, Nanjing 210009, China; 2Department of Medicinal Chemistry, School of Pharmacy, China Pharmaceutical University, Nanjing 210009, China

**Keywords:** Keap1-Nrf2, hydrogen sulfide, oxidative stress, antioxidant, anti-inflammation, ulcerative colitis

## Abstract

Ulcerative colitis (UC) is an idiopathic inflammatory disease of unknown etiology possibly associated with intestinal inflammation and oxidative stress. Molecular hybridization by combining two drug fragments to achieve a common pharmacological goal represents a novel strategy. The Kelch-like ECH-associated protein 1 (Keap1)-nuclear factor erythroid 2-related factor 2 (Nrf2) pathway provides an effective defense mechanism for UC therapy, and hydrogen sulfide (H_2_S) shows similar and relevant biological functions as well. In this work, a series of hybrid derivatives were synthesized by connecting an inhibitor of Keap1-Nrf2 protein–protein interaction with two well-established H_2_S-donor moieties, respectively, via an ester linker, to find a drug candidate more effective for the UC treatment. Subsequently, the cytoprotective effects of hybrids derivatives were investigated, and DDO-1901 was identified as a candidate showing the best efficacy and used for further investigation on therapeutic effect on dextran sulfate sodium (DSS)-induced colitis in vitro and in vivo. Experimental results indicated that DDO-1901 could effectively alleviate DSS-induced colitis by improving the defense against oxidative stress and reducing inflammation, more potent than parent drugs. Compared with either drug alone, such molecular hybridization may offer an attractive strategy for the treatment of multifactorial inflammatory disease.

## 1. Introduction

Ulcerative colitis (UC) is an idiopathic inflammatory bowel disease (IBD) with growing incidence seriously affecting millions of patients worldwide. Therapy for UC has been a significant medical challenge owing to the unknown exact etiology, poor prognosis, and high recurrence rate [1,2,3,4]. Although current medications including anti-inflammatory drugs, immunomodulatory agents, and hormones have certain curative effects to temporarily alleviate symptoms, their long-term clinical applications have been restricted by limited efficacy and serious adverse reactions [5,6,7]. Combination therapy of drugs with various pharmacological mechanisms is a treatment strategy for UC, while it is necessary to strike a difficult balance between safety and efficacy [8]. Several ongoing studies clearly show that this strategy is an opportunistic combination of available drugs, rather than based on hypotheses about potential additive or synergistic effects in mechanism [9,10]. Molecular hybridization represents a promising strategy for developing drugs with pleiotropic effects, particularly against a multifactorial disease of unknown etiology such as UC [11]. In this strategy, novel chemical entities combining two or more pharmacophoric moieties from different bioactive compounds are predicted to exert better synergism pharmacological efficacy, associated with the cooperative effects between individual parent drugs [12,13].

Hydrogen sulfide (H_2_S), an essential endogenous gaseous signaling molecule, has been demonstrated to be an important mediator of gastrointestinal mucosal defense and contribute to the repair of damaged tissues and promote the resolution of colitis [14,15,16,17,18,19,20,21]. Studies suggested that H_2_S exerts an anti-inflammatory effect by weakening the expression of many pro-inflammatory cytokines, as well as targeting NOD-like receptor family pyrin domain containing 3 (NLRP3) inflammasome in colitis [22,23,24,25]. In addition, H_2_S was reported to play an important role in maintaining the redox status by promoting scavenging of reactive oxygen species (ROS) [26,27,28], and it also modulates cysteine residues on key signaling molecules thereby promoting antioxidant mechanisms [29,30,31,32]. Small molecules that release H_2_S, often referred to as “H_2_S donors,” constitute not only useful investigative tools but also potential therapeutic agents. Various H_2_S donors, such as GYY4137, arylthiobenzamides, 4-hydroxythiobenzamide (4-OH-TBZ), 5-(4-hydroxyphenyl)-3H-1,2-dithiole-3-thione (ADT-OH) etc., have been developed and are already undergoing intensive investigation [33,34,35,36,37]. An established medicinal chemistry approach is related to the covalent incorporation of an H_2_S donor group into the structure of a biologically active drug to afford a novel hybrid compound [36,38,39]. Covalent linkage of H_2_S-donor together with nonsteroidal anti-inflammatory drugs (NSAIDs) can dramatically reduce gastrointestinal toxicity and enhance the therapeutical potency [40,41]. ATB-429, a hybrid linking H_2_S-releasing moiety ADT–OH via an ester bond with mesalamine, is more effective than mesalamine in reducing the severity of colitis, which is possibly ascribed to the anti-inflammatory effect of released H_2_S [42].

The Kelch-like ECH-related protein 1 (Keap1)-nuclear factor erythroid 2-related factor 2 (Nrf2) pathway is an important antioxidant defense mechanism to protect cells from oxidative stress and inflammation [43,44,45,46]. Nrf2 is a ubiquitously expressed transcription factor and regulates the expression of numerous anti-oxidative and anti-inflammatory genes through the binding of a specific cis-acting element known as the antioxidant response element (ARE) [47,48,49]. Previous studies showed that Nrf2 knockout in a UC mouse model results in severe colonic injury which may be due to excess production of ROS and inflammatory cytokines [50,51,52]. The Keap1-Nrf2 pathway is crucial in protecting intestinal integrity by diminishing inflammation responses and activating the antioxidant defense system. Therefore, activation of the Keap1-Nrf2 signaling pathway may be an effective therapeutic approach for UC [53,54,55,56,57,58,59,60].

DDO-1636, a potent Keap1-Nrf2 PPI inhibitor reported by our group previously which is bearing a carboxyl group crucial for the Keap1 binding, was chosen to be further modified [61]. The potential beneficial effects of targeting the Nrf2 pathway and H_2_S system in the treatment of colitis deserve investigation. Therefore, we have investigated the possibility that H_2_S could be used to enhance the efficacy of DDO-1636. Our research has led our group to develop novel H_2_S-releasing hybrids of DDO-1636, that demonstrate improved efficacy in models of colitis. A series of new hybrid molecules were designed and prepared by coupling DDO-1636 with H_2_S donor ADT-OH and 4-OH-TBZ derivatives via ester bonds, and polyethylene glycol was used as a linker owing to possessing biocompatibility. Here, we hypothesized that an H_2_S-releasing hybrid derivative of a Keap1-Nrf2 PPI inhibitor would exhibit enhanced effects relative to the parent drug alone. Further investigations were carried out to evaluate the therapeutic effects of the hybrid in dextran sodium sulfate (DSS)-induced experimental colitis in vitro and vivo.

## 2. Materials and Methods

### 2.1. Chemistry Section

#### 2.1.1. General Chemistry Methods

All chemicals purchased from commercial suppliers were used as received unless otherwise stated. All solvents were reagent grade and, when necessary, were purified and dried by standard methods. Reactions were monitored by thin-layer chromatography on silica gel plates (GF-254) visualized under UV light. Melting points were determined on a Mel-TEMP II melting point apparatus without correction. ^1^H NMR and ^13^CNMR spectra were recorded in CDCl_3_ or DMSO-*d_6_* on a Bruker Avance-300 instrument. Chemical shifts (δ) are reported in parts per million (ppm) from tetramethylsilane (TMS) using the residual solvent resonance (CDCl_3_: 7.26 ppm for ^1^H NMR, 77.16 ppm for ^13^C NMR; DMSO: 2.5 ppm for ^1^H NMR, 39.5 ppm for ^13^C NMR). Multiplicities are abbreviated as follows: s = singlet, d = doublet, t = triplet, q = quartet, and m = multiplet. HR-MS spectra were recorded on a Water Q-Tof micro mass spectrometer. Flash column chromatography was performed with 100–200 mesh silica gel, and yields refer to chromatographically and spectroscopically pure compounds. The purity of the compounds was analyzed on an LC-20AT prominence liquid chromatography system (Shimadzu, Japan) equipped with Wondasil C18 column (4.6 × 250 mm, 5 μm) using a mixture of solvent methanol/water at a flow rate of 1 mL/min and monitored by SPD-20A UV-detector at 254 nm. Yields and characterization, along with detailed assignments and copies of ^1^H and ^13^C NMR spectra are provided.

#### 2.1.2. Synthesis of DDO-1636

DDO-1636 was resynthesized according to the known procedure previously reported by our group [61]. White solid, yield: 83%. m.p. 225.1–227.0 °C; ^1^H NMR (300 MHz, DMSO-*d_6_*): δ 12.82 (s, 1H), 10.25 (s, 1H), 7.70 (s, 1H), 7.67 (s, 1H), 7.58 (s, 1H), 7.55 (d, *J* = 3.8 Hz, 2H), 7.53–7.50 (m, 1H), 7.10 (d, *J* = 2.2 Hz, 2H), 7.07 (d, *J* = 2.2 Hz, 2H), 7.04 (d, *J* = 1.9 Hz, 2H), 4.40 (s, 2H), 3.88 (s, 3H), 3.83 (s, 3H); HR-MS (ESI): found 579.0862 (C_26_H_24_N_2_O_8_S_2_ [M+Na]^+^ requires 579.0973); HPLC (80:20 methanol:water with 1‰ formic acid): t_R_ = 2.806 min, 96.330%.

#### 2.1.3. Synthesis of ADT-OH from Anethole Trithione

Anethole trithione (3.00 g, 12.48 mmol) was mixed with pyridine hydrochloride (12.42 g, 124.80mmol) and then was heated to 220 °C until the solids melted with nitrogen protection for about 1 h. After the temperature cooled to room temperature, 75 mL 1 M dilute hydrochloric acid was poured into the reaction and the resulting precipitate was filtered. The wet filter cake was recrystallized from ethanol to afford 2.70 g ADT-OH as an orange-red crystal. Yield: 55%. ^1^H NMR (300 MHz, DMSO-*d*_6_) δ 10.58 (s, 1H), 7.80 (d, *J* = 8.7 Hz, 2H), 7.72 (s, 1H), 6.97–6.86 (m, 2H); HR-MS (ESI): found 224.9511 (C_26_H_24_N_2_O_8_S_2_ [M−H]^−^ requires 224.9583); HPLC (80:20 methanol: water with 1‰ formic acid): t_R_ = 4.592 min, 95.708%.

#### 2.1.4. General Procedure for the Synthesis of Compounds 1b~1d and 2b~2d

ADT-OH (0.5 g, 2.21 mmol) or 4-OH-TBZ (0.34 g, 2.21 mmol) was dissolved in 20 mL anhydrous DMF. Then, K_2_CO_3_ (1.53 g, 11.05 mmol) and corresponding bromhydrin (6.63 mmol) were added to the solution. The reaction solution was heated to reflux at 65 °C for 5 h. Then, the mixture was filtered to remove insoluble salts and washed with dichloromethane three times. The filtrate was collected and concentrated in vacuo, and the crude product was purified by flash chromatography eluting to afford the targeted solid.

Compound 1b. Orange solid, yield: 97%. ^1^H NMR (300 MHz, DMSO-*d*_6_) δ 7.91 (d, J = 8.1 Hz, 2H), 7.81 (s, 1H), 7.12 (d, *J* = 8.2 Hz, 2H), 5.01 (s, 1H), 4.13 (s, 2H), 3.77 (s, 2H); ESI-*m/z*: 271.13 [M+H]^+^.

Compound 1c. Orange solid, yield: 72%. ^1^H NMR (300 MHz, DMSO-*d*_6_) δ 7.92 (s, 1H), 7.89 (s, 1H), 7.80 (s, 1H), 7.14 (d, *J* = 2.1 Hz, 1H), 7.11 (d, *J* = 2.2 Hz, 1H), 4.70–4.63 (m, 1H), 4.26–4.20 (m, 2H), 3.84–3.76 (m, 2H), 3.58–3.50 (m, 4H); ESI-*m/z*: 315.15 [M+H]^+^.

Compound 1d. Orange solid, yield: 78%. ^1^H NMR (300 MHz, DMSO-*d*_6_) δ 7.92 (s, 1H), 7.89 (s, 1H), 7.80 (s, 1H), 7.14 (s, 1H), 7.11 (s, 1H), 4.70–4.63 (m, 1H), 4.26–4.20 (m, 2H), 3.84–3.76 (m, 2H), 3.58–3.50 (m, 4H).; ESI-*m/z*: 359.14 [M+H]^+^.

Compound 2b. White solid, yield: 71%. ^1^H NMR (300 MHz, DMSO-*d*_6_) δ 10.03 (s, 1H), 9.56 (s, 1H), 7.83–7.75 (m, 2H), 7.18–7.10 (m, 2H), 4.96 (t, *J* = 5.5 Hz, 1H), 4.11 (m, *J* = 5.4, 4.3 Hz, 2H), 3.80–3.72 (m, 2H); ESI-*m/z*: 198.22 [M+H]^+^.

Compound 2c. White solid, yield: 74%. ^1^H NMR (300 MHz, DMSO-*d*_6_) δ 10.13 (s, 1H), 9.64 (s, 1H), 7.82–7.76 (m, 2H), 7.18–7.12 (m, 2H), 4.70–4.63 (m, 1H), 4.25–4.18 (m, 2H), 3.82–3.75 (m, 2H), 3.53 (q, *J* = 3.2, 2.6 Hz, 4H); ESI-*m/z*: 242.02 [M+H]^+^.

Compound 2d. White solid, yield: 73%. ^1^H-NMR (300 MHz, DMSO-*d*_6_): δ 9.94 (s, 1H), 9.64 (s, 1H), 7.82–7.76 (m, 2H), 7.18–7.12 (m, 2H), 4.70–4.63 (m, 1H), 4.25–4.18 (m, 2H), 3.82–3.75 (m, 2H), 3.53 (dd, *J* = 6.5, 4.0 Hz, 4H), 3.37 (m, 4H); ESI-*m/z*: 286.33 [M+H]^+^.

#### 2.1.5. General Procedure for the Synthesis of Compounds DDO-1901~DDO-1908

A solution of DDO-1636 (0.19 g, 0.34 mmol), DCC (0.12 g, 0.56 mmol), and DMAP (0.03 g, 0.28 mmol) in anhydrous THF (5 mL) was stirred at room temperature for 30 min. Then, 1a~1d and 2a~2d (0.28 mmol) were added to the solution and stirred at room temperature for about 8 h until esterification was complete. The mixture was poured into 10 mL of H_2_O, and extracted with ethyl acetate (10 mL × 3). The organic layers were combined, washed with saturated NaCl solution, dried over anhydrous Na_2_SO_4_, and concentrated in vacuo. The residues were purified by chromatography on silica gel to afford DDO-1901~DDO-1908.

Compound DDO-1901. Orange solid, yield: 71%. m.p. 128.7–130.1 °C; ^1^H NMR (300 MHz, DMSO-*d*_6_) δ 10.28 (s, 1H), 8.20–8.13 (m, 2H), 8.00–7.95 (m, 2H), 7.84 (s, 1H), 7.67 (m, J = 18.5, 8.9 Hz, 4H), 7.58–7.52 (m, 2H), 7.25–7.21 (m, 2H), 7.13 (d, *J* = 2.1 Hz, 1H), 7.09 (t, *J* = 2.6 Hz, 4H), 7.06 (d, *J* = 2.3 Hz, 1H), 4.85 (d, *J* = 4.3 Hz, 2H), 3.88 (s, 3H), 3.82 (s, 3H); ^13^C NMR (75 MHz, Chloroform-*d*) δ 215.45, 171.44, 167.25, 163.43, 163.25, 152.95, 136.10, 134.24, 133.01, 132.60, 130.59, 130.25, 129.54, 129.51, 129.23, 128.78, 128.26, 127.75, 127.53, 127.34, 124.37, 122.63, 121.76, 120.28, 116.58, 114.26, 114.10, 55.69, 55.64, 53.39; HR-MS (ESI): found 765.0516 (C_35_H_28_N_2_O_8_S_5_ [M+H]^+^ requires 765.0539); HPLC (80:20 methanol:water): t_R_ = 5.318 min, 98.231%.

Compound DDO-1902. Orange solid, yield: 74%. m.p. 104.5–105.2 °C; ^1^H NMR (300 MHz, DMSO-*d*_6_) δ 10.24 (s, 1H), 8.12 (m, *J* = 9.5, 7.0, 1.5 Hz, 2H), 7.89 (d, *J* = 2.0 Hz, 1H), 7.86 (d, *J* = 2.2 Hz, 1H), 7.80 (s, 1H), 7.68 (d, *J* = 2.0 Hz, 1H), 7.66 (d, *J* = 2.1 Hz, 1H), 7.57 (s, 1H), 7.55 (d, *J* = 2.1 Hz, 1H), 7.51–7.46 (m, 2H), 7.08–7.02 (m, 6H), 7.00 (d, *J* = 1.8 Hz, 2H), 4.65–4.46 (m, 2H), 4.39–4.34 (m, 2H), 4.21 (q, *J* = 4.2, 3.8 Hz, 2H), 3.85 (s, 3H), 3.81 (s, 3H); ^13^C-NMR (75 MHz, DMSO-*d*_6_): δ 215.31, 174.16, 169.10, 163.35, 162.89, 161.91, 134.77, 134.62, 134.01, 132.96, 131.98, 130.41, 130.23, 129.67, 129.47, 126.81, 124.83, 124.43, 123.78, 121.49, 115.91, 114.77, 56.12; HR-MS (ESI): found 809.0706 (C_37_H_32_N_2_O_9_S_5_ [M+H]^+^ requires 809.0695); HPLC (80:20 methanol:water): t_R_ = 5.296 min, 94.175%.

Compound DDO-1903. Orange solid, yield: 71%. m.p. 87–88 °C; ^1^H NMR (300 MHz, DMSO-*d*_6_) δ 10.26 (s, 1H), 8.19–8.08 (m, 2H), 7.90–7.84 (m, 2H), 7.79 (s, 1H), 7.71–7.64 (m, 2H), 7.60–7.50 (m, 4H), 7.10–7.01 (m, 8H), 4.62–4.43 (m, 2H), 4.15 (dq, *J* = 10.3, 4.8, 4.2 Hz, 4H), 3.87 (s, 3H), 3.81 (s, 3H), 3.77–3.69 (m, 2H), 3.60 (t, *J* = 4.7 Hz, 2H); ^13^C-NMR (75 MHz, DMSO-*d*_6_): δ 215.25, 174.25, 169.10, 163.34, 162.86, 162.33, 134.64, 134.04, 132.96, 132.04, 130.39, 130.23, 129.72, 129.44, 127.47, 127.13, 126.80, 124.86, 124.18, 123.79, 121.53, 115.93, 114.77, 68.00, 60.25, 56.19, 56.10; HR-MS (ESI): found 853.0966 (C_39_H_36_N_2_O_10_S_5_ [M+H]^+^ requires 853.0945); HPLC (80:20 methanol:water): t_R_ = 5.228 min, 95.191%.

Compound DDO-1904. Orange solid, yield: 72%. m.p. 72.3–73.4 °C; ^1^H NMR (300 MHz, DMSO-*d*_6_) δ 10.26 (s, 1H), 8.18–8.09 (m, 2H), 7.91–7.85 (m, 2H), 7.80 (s, 1H), 7.68 (m, *J* = 8.9, 1.4 Hz, 2H), 7.59 (d, *J* = 2.8 Hz, 1H), 7.57–7.51 (m, 3H), 7.11–6.99 (m, 8H), 4.63–4.43 (m, 2H), 4.18 (m, *J* = 5.6, 3.5 Hz, 2H), 4.11 (q, *J* = 4.2 Hz, 2H), 3.88 (d, *J* = 3.3 Hz, 3H), 3.83 (d, *J* = 3.1 Hz, 3H), 3.79–3.71 (m, 2H), 3.59–3.50 (m, 6H); ^13^C-NMR (75 MHz, DMSO-*d*_6_): δ 215.25, 174.27, 169.10, 163.34, 162.86, 162.39, 134.62, 134.11, 132.97, 132.06, 130.40, 129.72, 129.45, 127.44, 124.85, 124.15, 123.80, 120.50, 115.94, 114.77, 70.29, 69.16, 68.45, 68.07, 56.19, 56.10; HR-MS (ESI): found 897.1291 (C_41_H_40_N_2_O_11_S_5_ [M+H]^+^ requires 897.1235); HPLC (80:20 methanol:water): t_R_ = 5.210 min, 97.897%.

Compound DDO-1905. White solid, yield: 71%. m.p. 212.3–213.5 °C; ^1^H NMR (300 MHz, DMSO-*d*_6_) δ 10.28 (s, 1H), 9.94 (s, 1H), 9.55 (s, 1H), 8.16 (m, *J* = 9.8, 7.7, 4.1 Hz, 2H), 7.98–7.87 (m, 2H), 7.73–7.60 (m, 4H), 7.58–7.50 (m, 2H), 7.15–7.03 (m, 8H), 4.83 (d, *J* = 2.1 Hz, 2H), 3.88 (s, 3H), 3.82 (s, 3H); ^13^C-NMR (75 MHz, DMSO-*d*_6_): δ 199.43, 168.00, 163.49, 162.91, 152.63, 137.83, 134.71, 134.10, 132.96, 131.96, 130.49, 130.32, 129.47, 129.28, 127.43, 127.25, 126.90, 124.74, 123.89, 121.73, 121.39, 114.90, 114.80, 56.23, 56.12, 53.84; HR-MS (ESI): found 692.1191 (C_33_H_29_N_3_O_8_S_3_ [M+H]^+^ requires 692.1114); HPLC (80:20 methanol:water): t_R_ = 4.225 min, 96.476%.

Compound DDO-1906. White solid, yield: 72%. m.p. 208.1–209.3 °C; ^1^H NMR (300 MHz, DMSO-*d*_6_) δ 10.25 (s, 1H), 9.95 (s, 1H), 9.64 (s, 1H), 8.13 (t, *J* = 8.9 Hz, 2H), 7.78 (d, *J* = 7.9 Hz, 2H), 7.67 (d, *J* = 8.4 Hz, 2H), 7.56 (d, *J* = 8.2 Hz, 2H), 7.49 (d, *J* = 3.8 Hz, 2H), 7.15–6.93 (m, 8H), 4.55 (d, J = 12.8 Hz, 2H), 4.36 (s, 2H), 4.21 (s, 2H), 3.84 (d, *J* = 10.3 Hz, 6H); ^13^C-NMR (75 MHz, DMSO-*d*_6_): δ 169.09, 163.35, 162.89, 161.94, 134.67, 134.02, 132.97, 131.97, 130.41, 129.65, 129.47, 127.34, 127.09, 126.80, 124.84, 123.81, 121.54, 119.57, 115.93, 114.77, 103.63, 66.48, 63.52, 56.16, 56.10, 53.62; HR-MS (ESI): found 736.1382 (C_35_H_23_N_3_O_9_S_3_ [M+H]^+^ requires 736.1337); HPLC (80:20 methanol:water): t_R_ = 4.053 min, 96.726%.

Compound DDO-1907. White solid, yield: 74%. m.p. 206.5–207.6 °C; ^1^H-NMR (300 MHz, DMSO-*d_6_*): δ 10.25 (s, 1H), 9.95 (s, 1H), 9.64 (s, 1H), 8.13 (t, *J* = 8.9 Hz, 2H), 7.78 (d, *J* = 7.9 Hz, 2H), 7.67 (d, *J* = 8.4 Hz, 2H), 7.56 (d, *J* = 8.2 Hz, 2H), 7.49 (d, *J* = 3.8 Hz, 2H), 7.15–6.93 (m, 8H), 4.55 (d, *J* = 12.8 Hz, 2H), 4.36 (s, 2H), 4.21 (s, 2H), 3.84 (d, *J* = 10.3 Hz, 6H), 3.61–3.52 (m, 4H); HR-MS (ESI): found 780.1681 (C_37_H_37_N_3_O_10_S_3_ [M+H]^+^ requires 780.1641); HPLC (80:20 methanol:water): t_R_ = 3.759 min, 99.038%.

Compound DDO-1908. White solid, yield: 75%. m.p. 203.7–204.9 °C; ^1^H-NMR (300 MHz, DMSO-*d_6_*): δ 10.23 (s, 1H), 9.96 (s, 1H), 9.64 (s, 1H), 8.13 (t, *J* = 8.9 Hz, 2H), 7.78 (d, *J* = 7.9 Hz, 2H), 7.67 (d, *J* = 8.4 Hz, 2H), 7.56 (d, *J* = 8.2 Hz, 2H), 7.49 (d, *J* = 3.8 Hz, 2H), 7.15–6.96 (m, 8H), 4.55 (d, *J* = 12.8 Hz, 2H), 4.36 (s, 2H), 4.21 (s, 2H), 3.84 (d, *J* = 10.3 Hz, 6H), 3.61–3.52 (m, 4H), 3.46–3.40 (m, 4H); HR-MS (ESI): found 824.1932 (C_39_H_41_N_3_O_11_S_3_ [M+H]^+^ requires 824.1903); HPLC (80:20 methanol:water): t_R_ = 3.657 min, 95.644%.

### 2.2. Hydrogen Sulfide Release Evaluation (Methylene Blue Assay)

Na_2_S was dissolved in phosphate-buffered solution (pH 7.4) in a 100 mL volumetric flask, which was used as the stock solution (5 mM), and then standard solutions of 5, 10, 20, 40, 60, 80, 100, and 150 μM in 50 mL volumetric flask were prepared. A total of 200 μL of each standard solution was taken into a 1.5 mL eppendorf tube, and then 200 μL zinc acetate (1%, *w*/*v*), 600μL *N*,*N*-dimethyl-1,4-phenylenediaminesulfate (0.2% *w/v* in 20%H_2_SO_4_ solution), and 50 μL ferric chloride (10% *w/v* in 0.2% H_2_SO_4_ solution) were added and stored at room temperature for 20 min (each reaction was carried out in triplicate). The absorbance of each resulting mixture was recorded at 670 nm in a UV–Vis spectrophotometer (Shimadzu, UV-1900, Kyoto, Japan) to draft the Na_2_S calibration curve. Compounds were dissolved in DMSO as solutions (10 mM) and stored at −20 °C; each compound (final conc.: 200μM) was dissolved in phosphate-buffered solution (pH 7.4) containing 1 mM TCEP. The resultant solution was sealed and incubated at 37 °C. At different time points, 200 μL of the reaction solution was taken into a 1.5 mL eppendorf tube, and then the experiment proceeded as the method described above. Absorbance was measured at 670 nm after 20 min, and samples were assayed in triplicate. According to the absorbance of each sample, the H_2_S concentration was calculated based on a calibration curve of Na_2_S.

### 2.3. Esterase Triggering DDO-1636 Release as Monitored by HPLC

DDO-1901 (final Conc. 200 μM) was added to PBS (10 mL) with 1 unit/mL esterase followed by vortex mixing. The solution was incubated at 37 °C and conducted in triplicate. A total of 200 μL reaction mixture was taken into a 1.5 mL eppendorf tube at appropriate time intervals. Protein was precipitated by adding 200 μL methanol, and samples were subjected to vortex mixing and then centrifugation for 5 min at 12,000 rpm to deproteinize. The resulting supernatants were withdrawn and analyzed by HPLC. Peak areas were recorded to calculate the percentage of compounds. Shimadzu LC-20AT prominence liquid chromatography system equipped with an SPD-20A UV-detector: Wondasil C18 column (4.6 × 250 mm, 5 μm); Mobile phase: methanol 80%; Flow rate: 1 mL/min. A standard curve for compounds was made to fit the measured concentrations.

### 2.4. Cell Culture

Human NCM460 colonocytes (INCELL, San Antonio, TX, USA) were cultured in Dulbecco’s modified Eagle medium (DMEM) (Gibco™, 11995065, Thermo Fisher Scientific, Waltham, MA, USA) supplemented with 10% (*v*/*v*) fetal bovine serum (FBS) (ExCell Bio, FSP500) and penicillin/streptomycin, in a humidified atmosphere of 5% CO_2_ and 95% air at 37 °C.

### 2.5. Fluorescence Probe Studies of H_2_S Release in Cells

A H_2_S fluorescent probe WSP-5 (MKBio, Shanghai, China) was used for the detection of H_2_S release. The NCM460 cells were seeded in a 12-well plate one day before the experiment. Compounds or other control compounds were dissolved to culture medium to obtain a final concentration of 50 μM and then added into NCM460 cells. The cells were then incubated with the compound at 37 °C with 5% CO_2_. After incubating for 45 min, the medium was removed, and cells were washed three times with 1× PBS. Then, the cells were incubated with WSP-5 (50 μM) in PBS at 37 °C for 20 min in the dark. After the PBS was removed and cells were washed, analysis was done with a fluorescence microscope (OLYMPUS DP72, Tokyo, Japan) equipped with a U-RFL-T power supply. The magnification is 20×.

### 2.6. Cell Viability Study

The cell viability was tested by the MTT assay. NCM460 cells were plated in 96-well plates at a density of 1 × 10^4^ cells/mL and allowed to attach overnight. Solutions of compounds were applied in the medium for incubation at 37 °C under a humidified atmosphere containing 5% CO_2_ for 24 h. MTT (0.5 mg/mL) was added, and the cells were incubated for another 4 h. Medium/MTT solutions were removed carefully by aspiration; the MTT formazan crystals were dissolved in 150 mL of DMSO; and absorbance was determined at 570 nm on Molecular Devices SpectraMax i3x Reader after shaking gently for 3 min. For each independent experiment, the assays were performed in three replicates.

### 2.7. Western Blotting

Anti-Nrf2 (ab62352), anti-HO-1 (ab52947), and anti-NLRP3 (ab263899) antibodies were purchased from Abcam (Abcam Technology, England). Anti- Caspase 1 p20 (PA5-99390) antibody was purchased from Invitrogen (Thermo Fisher Scientific, Waltham, MA, USA). Anti-α-Tubulin (66031-1-Ig), anti-NQO1(67240-1-Ig), and anti-GCLM (14241-1-AP) antibodies were purchased from Proteintech (Proteintech Group, Rosemont, IL, USA). NCM460 cells were incubated with indicated compounds for the specified time interval, and cells were harvested and lysed using RIPA lysis buffer with Roche protease inhibitor complete cocktail. The solution was centrifuged for 10 min at 4 °C, and the total protein concentrations were determined by BCA protein assay. Individual cell lysates were separated by sodium dodecyl sulfate-polyacrylamide gel electrophoresis (12% gel, SDS-PAGE), and the separated proteins were transferred to PVDF membranes by wet transfer. The membrane was blocked in 5% fat-free milk, followed by incubation with a primary antibody overnight at 4 °C and a horseradish peroxidase (HRP)-conjugated secondary antibody for 1 h at room temperature. The membrane was imaged by Tanon 5200 Multi Imaging Workstation.

### 2.8. Detection of SOD, GPx, and MDA Activities and the Ratio of GSH/GSSG

The activities of SOD (Total Superoxide Dismutase Assay Kit with WST-8, S0101S, Beyotime, Shanghai, China), GPx (Total Glutathione Peroxidase Assay Kit with NADPH, S0058, Beyotime, Shanghai, China), and MDA (Lipid Peroxidation MDA Assay Kit, S0131S, Beyotime, Shanghai, China) were determined using the corresponding detection kits according to the manufacturer’s instructions. The ratio of GSH/GSSG was evaluated using a commercially available kit according to the manufacturer’s instructions (GSH and GSSG Assay Kit, S0053, Beyotime, Shanghai, China).

### 2.9. IL-6, IL-1β, and TNF-α Production

Levels of IL-6 (Human IL-6 ELISA Kit, EK0410, Boster, Wuhan, China), IL-1β (Human IL-1 beta ELISA Kit, EK0392, Boster, Wuhan, China), and TNF-α (Human TNFα ELISA Kit, EK0525, Boster, Wuhan, China) in human NCM460 cell culture supernatant were evaluated using commercially available kits according to the manufacturer’s instruction.

### 2.10. Animal Experiments

#### 2.10.1. Animal

Animal studies were conducted according to protocols approved by the Institutional Animal Care and Use Committee of China Pharmaceutical University, and the protocol code is 2022-12-009. All animals were appropriately used in a scientifically valid and ethical manner. Male C57BL/6 mice (6–8 weeks old, 17–20 g) (Gempharmatech, Nanjing, China) were acclimatized under a temperature- and light/dark cycle-controlled environment and 60% humidity for 1 week before the experiments and fed with a standard laboratory rodent diet and water. Mice were given free access to diet and water during the experiment.

#### 2.10.2. DSS-induced Acute Colitis

DSS (molecular weight 36–50 kDa) (MP Biomedicals, Morgan Irvine, CA, USA) was dissolved in distilled water to a concentration of 3%. Mice were challenged with 3% DSS in drinking water for 7 days to induce colitis and treated daily with different drugs by direct intraperitoneal injection. Thirty-six mice were divided into six groups at random (6 mice per group): (1) Control group (drinking water), (2) DSS model group (3% *w*/*v*), (3) DSS + DDO-1901 group (40 mg/kg), (4) DSS + DDO-1901 (10 mg/kg), (5) DSS + DDO-1636 group (40 mg/kg), and (6) DSS + ADT-OH group (40 mg/kg). Mice in the normal group drank normal water every day. The model group drank water mixed with 3% DSS. During the DSS and drug treatment, animals were inspected daily in the morning, and body weight and food/fluid consumption, as well as scores for diarrhea and bleeding, were recorded. The sum of the scores for diarrhea, bleeding, and body weight loss was used as a disease activity index (DAI). At the end of the period, the mice were sacrificed, and blood samples were collected for the subsequent analysis, and their colons were removed and measured for their length. The colons were fixed in 10% buffered formalin (pH 7.4) for at least 24 h for further histopathological assessment and study.

#### 2.10.3. IL-6, IL-1β, and TNF-α Production

Freshly collected blood samples were kept at room temperature for 2 h and centrifuged at 3000× *g* for 10 min at 4 °C, and the supernatant was used for detection and analysis. The secretion of IL-6, IL-1β, and TNF-α in mouse serum samples was measured by double-antibody sandwich ELISA according to the manufacturer’s instructions of these commercially available kits (Mouse IL-1 beta ELISA Kit, EK0394, Boster), IL-6 (Mouse IL-6 ELISA Kit, EK0411, Boster, Wuhan, China), and TNF-α (Mouse TNFα ELISA Kit, EK0527, Boster, Wuhan, China).

#### 2.10.4. Histopathological Examination

Specimens of the colon fixed with 10% buffered formalin were embedded in paraffin. Each section (4 μm) was stained with hematoxylin and eosin (H&E) staining. The fixed sections were examined by light microscopy for the presence of lesions. Histological evaluation of the severity of inflammation was performed using a scoring system by a pathologist who was blinded to the treatment.

## 3. Results and Discussion

### 3.1. Chemistry

The overall synthetic route is outlined in Figure 1. The synthetic procedures for the preparation of compounds DDO-1901~DDO-1904 and DDO-1905~DDO-1908 were based on the condensation of DDO-1636 with ADT-OH or 4-OH-TBZ, respectively, into the corresponding esters. H_2_S-donor 4-OH-TBZ is commercially available; ADT-OH was synthesized starting from anethole trithione, which was subsequently demethylated upon heating with pyridine hydrochloride to afford ADT-OH. DDO-1636 was synthesized according to our previous report. Firstly, ADT-OH and 4-OH-TBZ were treated with several bromhydrins, respectively, to obtain the H_2_S donating derivatives by substitution reactions. The intermediates were subjected to a condensation reaction between the DDO-1636 carboxylic function and the hydroxyl functional group, via DCC in the presence of DMAP, thus obtaining the final compounds DDO-1901-DDO-1908.

### 3.2. Hybrid Compounds Are Capable of Releasing H_2_S

To evaluate the ability of the H_2_S-donors and the novel synthesized hybrid molecules to generate H_2_S, the methylene blue assay was proceeded in phosphate-buffered solution (pH 7.4) in the presence of TCEP (a water-soluble effective mercaptan), which was used as an accelerator. Representative release curves about the H_2_S concentration to time were summarized in Figure 1A,B, showing that the generation of H_2_S elevated as time increased and reached the peak value at 30 or 40 min. All the tested compounds could release H_2_S in a relatively slow manner, albeit with different features in the quantitative aspects which may be related to their chemical structures. Comparing the results obtained for the two different series of derivatives, we evidenced that when 4-OH-TBZ was used as H_2_S-donor, the amount of produced H_2_S was relatively lower. In addition, the length of the linker also seemed to be an influencing factor in the release of H_2_S. With the increased length of chains, lower H_2_S concentration was observed compared with the parent compound. Possibly, this result is caused by linkers, as the big steric hindrance disturbs H_2_S release. The best result was obtained with compound DDO-1901 that produced a relatively consistent of H_2_S amount compared to the free H_2_S-donor moiety ADT-OH, indicating that the introduction of DDO-1636 moiety did not affect the H_2_S release.

Having confirmed H_2_S release from hybrid molecules in PBS buffer, we next investigated whether these compounds could generate H_2_S in live cells, and WSP-5, an H_2_S-selective fluorescent probe, was used to monitor H_2_S accumulation in NCM460 cells. As shown in Figure 1C, the NCM460 cells displayed negligible H_2_S-derived fluorescence in the absence of compounds, while the addition of compounds resulted in a remarkable fluorescent signal, demonstrating that these hybrids can successfully generate H_2_S in the cells. Moreover, the fluorescence intensity of the derivatives decreased compared with the parent H_2_S donor, and the longer the chains of the fragment, the lower the concentration of H_2_S-releasing, which is consistent with the H_2_S release properties in PBS buffer. Additionally, then, DDO-1901 was selected for further evaluation owing to the maximum reservation of H_2_S release property of ADT-OH. Additional HPLC studies showed that DDO-1901 could generate DDO-1636 within 5 h after incubation with esterase at 37 °C (conversion rate > 98%) (Appendix A).

### 3.3. DDO-1901 Protects NCM460 Cells from DSS-Induced Injury

To establish a UC model in cells, dextran sulfate sodium salt (DSS), a common inflammation-inducing agent, was used to induce injury in NCM460 cells. MTT assay was conducted to evaluate the protective effects of the hybrids against the DSS-induced cell damage. After the treatment with various concentrations of DSS, NCM460 cell viability decreased significantly, which was reduced below 50% when the concentration of DSS was higher than 20 mg/mL (Appendix A). Next, NCM460 cells were pretreated with DDO-1901~DDO-1908 (10 μM) for 24 h, followed by the addition of DSS (20 mg/mL) treatment for another 12 h. As shown in Figure 2A, all the compounds exhibited cytoprotective effects against DSS-induced injury, and DDO-1901 performed the optimal activity. Further studies showed that DDO-1901 antagonized the cell damage caused by DSS in the time-dependent manner (Figure 2B), and cell viability was increased significantly with the pretreatment of various concentrations of DDO-1901 (Figure 2C). In addition, DDO-1901 exhibited a superior cytoprotective effect against the DSS-induced cell damage than ADT-OH and DDO-1636 (Figure 2D), indicating that the synergistic protective effect might have been produced between H_2_S donor ADT-OH and Keap1-Nrf2 PPI inhibitor DDO-1636.

### 3.4. DDO-1901 Activates Nrf2-regulated Antioxidant System in NCM460 Cells

The Keap1-Nrf2 pathway has been reported to regulate several detoxification enzymes and antioxidant proteins including NAD(P)H: quinone oxidoreductase 1 (NQO1), glutamate-cysteine ligase modifier subunit (GCLM), and heme oxygenase-1 (HO-1), which are essential for reducing the risk of intestinal inflammation through the cellular defense system [62,63,64,65]. As mentioned above, Keap1-Nrf2 inhibitor DDO-1636 can activate the Nrf2-regulated cytoprotective defense system, and H_2_S donors are also reported to increase the expression of Nrf2 and its subsequent antioxidant proteins to activate the antioxidant effect. Therefore, in order to identify the effect of the hybrid compound DDO-1901 on the regulation of the Nrf2 pathway, Western blot experiments were performed in NCM460 cells. It was found that DDO-1901 had a concentration-dependent impact on Nrf2, NQO1, GCLM, and HO-1 protein expression levels that were more potent than with the parental compounds ADT-OH and DDO-1636, respectively (Figure 3A,B).

Subsequently, in order to investigate the impact of DDO-1901 on antioxidant capacity under inflammatory conditions, the activities of representative enzymes such as superoxide dismutase (SOD) and glutathione peroxidase (GPx) were determined. Pretreatment with DDO-1901 significantly restored the activities of SOD and GPx decreased by DSS (20 mg/mL), enhancing the antioxidant capacity of NCM460 cells to protect against DSS-induced oxidative damage (Figure 3C,D). We further measured the GSH/GSSG ratio, an important indicator of the glutathione (GSH)-based antioxidant system. Treatment with DSS remarkably caused the decline of the GSH/GSSG ratio, and pretreatment of DDO-1901 significantly restored the ratio nearly back to normal and promoted the synthesis of antioxidant GSH (Figure 3E). DSS (20 mg/mL) exposure increased malondialdehyde (MDA), an endogenous genotoxic product of lipid peroxidation (Figure 3F), while pretreatment of DDO-1901 effectively reduced the MDA level. The above study indicated that DDO-1901 could protect NCM460 cells against DSS-induced colon oxidative injury through activating the Nrf2-mediated antioxidant pathway.

### 3.5. DDO-1901 Represses DSS-Induced NLRP3 Inflammasome Activation and Pro-Inflammatory Cytokines Production in NCM460 Cells

In the DSS-induced colitis model, evidence reveals that DSS can directly stimulate NLRP3 inflammasome activation, which resulted in the initiation of severe intestinal inflammation [66,67,68]. The stimulated NLRP3 is united with apoptosis associated speck-like protein (ASC), which in turn recruits caspase-1 that can be cleaved to its activated form, subsequently promoting the maturation and secretion of pro-inflammatory cytokines to participate in the development of various inflammatory diseases [69,70]. It is reported that H_2_S exerts an anti-inflammatory effect on DSS-induced colitis by downregulating cleaved caspase-1 (p20) and NLRP3 expression [22,71]. Activation of Nrf2 was also confirmed to play a key anti-inflammatory role by significantly inhibiting NLRP3 inflammasome activation [63,72,73,74]. We aim to investigate the effects of novel hybrid DDO-1901 on DSS-induced NLRP3 inflammasome activation and inflammatory cytokines production in vitro. We first detected the expression of NLRP3 inflammasome related proteins in NCM460 cells by Western blotting analysis. DDO-1901 significantly inhibited the DSS-elevated protein level of NLRP3 and caspase-1 (p20), more potent than the parent drug ADT-OH and DDO-1636 (Figure 4A,B). Furthermore, unrestrained pro-inflammatory cytokines production has a great influence on the pathogenesis of DSS-induced colitis. The ELISA results demonstrated that the levels of pro-inflammatory cytokines IL-6, IL-1β, and TNF-α were meaningfully increased in DSS-induced NCM460 cells (Figure 4C–E), and the elevation of these factors could be dramatically reversed close to normal levels by DDO-1901, alleviating inflammatory responses in DSS-induced NCM460 cells.

### 3.6. DDO-1901 Alleviates the Pathological Symptoms of DSS-Induced Colitis in Mice

Considering the potent antioxidant and anti-inflammatory impacts of DDO-1901 exhibited in DSS-induced colitis in NCM460 cells, we hypothesized that DDO-1901 can play a therapeutic potential in DSS-induced colitis in mice, a well-established preclinical model exhibiting a majority of phenotypic features associated with human inflammatory bowel disease. The mice model was established via feeding with 3% DSS solution for consecutive 7 days, and the body weight, colon length, and disease activity index (DAI) were analyzed to evaluate the severity extent of colitis. DAI is a combined score with the incidence of weight loss, diarrhea, and rectal bleeding, which was then calculated according to a standard scoring system. As shown in Figure 5A–D, compared with the control group, mice in the DSS group developed serious symptoms of colitis including decreased body weight, diarrhea, hematochezia, and shortened colon lengths, exerting higher DAI scores. According to the data, administration of DDO-1901 showed remarkable therapeutic effects, more potent than ADT-OH and DDO-1636 at the same dose (40 mg/kg). DDO-1901 ameliorated weight loss and elevated DAI score induced by DSS, and colon length was significantly recovered from DSS damage in DDO-1901 injected groups.

To further confirm the effect of DDO-1901 on tissue inflammation and injury, colonic sections of different groups were histologically stained and examined. Hematoxylin and eosin (H&E) staining (Figure 5E) was performed, and the inflammatory cell infiltration and tissue damage of DSS-induced colitis were translated into a histological colitis severity score. Histopathological analysis (Figure 5F) revealed that DSS elicited colonic inflammation including loss of epithelial crypts, disruption of the colonic mucosa, and infiltration of inflammatory cells, directly causing higher histological scores. By contrast, DDO-1901 improved the pathological changes and decreased the histological scores of DSS mice, and the disorganized colonic mucosa was turned into well-organized colonic architecture with much less inflammatory cell infiltration and tissue damage. All the evidence above indicated that DDO-1901 administration significantly ameliorates pathological symptoms of DSS-induced colitis in vivo.

### 3.7. DDO-1901 Enhances the Antioxidant Defenses via Activation of Nrf2 against DSS-Induced Colitis in Mice

Next, Western blot analysis was conducted to measure the protein levels of Nrf2 and its downstream proteins NQO1, GCLM, and HO-1 to evaluate whether DDO-1901 was able to activate the Nrf2 antioxidant signaling pathway in colon tissues. Compared with the blank control group, Nrf2 was slightly upregulated in the DSS-treated group, which may be caused by the self-preservation mechanism in vivo. Consistently, DDO-1901 was capable of further enhancing the expression of Nrf2 than ADT-OH and DDO-1636, and the NQO1, GCLM, and HO-1 levels were also increased (Figure 6A).

Antioxidant enzymes SOD and GPx and lipid peroxidation indicator MDA are important parameters of oxidative stress in acute colitis. As displayed in Figure 6B–D, a sharp decrease of SOD and Gpx activities and an increase of MDA content were observed in colon homogenate of DSS-treated mice, while such reduced antioxidant capacity was all dramatically reversed by DDO-1901 (40 mg/kg), enhancing the antioxidant defense system to attenuate DSS-induced colitis.

### 3.8. DDO-1901 Relieves the Inflammation Conditions in the DSS-Induced Colitis in Mice

To determine whether DDO-1901 alleviated inflammation in DSS-induced colitis, colon tissues of different experimental groups were harvested to assess the expression levels of NLRP3 and caspase-1. It was found that DDO-1901 reduced the expression of NLRP3 as well as caspase-1 in colon tissues compared to those treated with DSS only (Figure 7A). A complex range of inflammatory signal pathway processes accelerates the generation of pro-inflammatory cytokines, closely related to exacerbated intestinal inflammation. To investigate the expression of cytokines in DSS-induced colitis mice, serum was extracted, and levels of pro-inflammatory cytokines (IL-6, IL-1β, and TNF-α) were determined. As delineated in Figure 7B–D, after induction with DSS, the level of cytokines was significantly higher compared with the control groups. DDO-1901 is more potent in preventing DSS-induced secretion of pro-inflammatory cytokines in the serum of mice thus rescuing them from inflammatory injury.

## 4. Conclusions

In summary, hybrid compounds combining Keap1-Nrf2 inhibitor DDO-1636 and H_2_S donors have been designed and synthesized as novel therapeutic agents. On account of DSS-induced intestinal inflammation being similar to the pathological characterization of human inflammatory bowel disease, the DSS-induced colitis model was established in this study. All the hybrid compounds exhibited an increased cytoprotective effect against DSS-induced injury in NCM460 cells. In this aspect, DDO-1901, which exerted the most remarkable effect, was proved to ameliorate DSS-induced colitis against oxidative stress by activating the Nrf2 pathway. As a derivate derived from DDO-1636 and H_2_S donor ADT-OH, DDO-1901 possessed great anti-inflammatory activities and could suppress the production of pro-inflammatory cytokines and enhance antioxidant defense. In addition, further studies in vivo showed that DDO-1901 could more efficiently alleviate the intestinal injury and symptoms of DSS-induced colitis mice than treatment with DDO-1936 and ADT-OH alone, including body weight loss, colon length shortening, DAI score decrease, and serious histopathological changes. Further investigation showed that DDO-1901 protected the colon against damage from oxidative stress by up-regulating expressions of Nrf2 and downstream antioxidant proteins, increasing antioxidative enzymes activities, and reducing MDA content in colitis mice. All these results indicated that compared to the treatment with either DDO-1636 or ADT-OH individually, DDO-1901 exhibited synergistic effects by significantly inhibiting inflammation and oxidative stress to alleviate colitis. Overall, the synergic combination of Keap1-Nrf2 inhibitors and H_2_S donors has the potential for ulcerative colitis treatment, and molecular hybridization may be a promising strategy for the therapy of multifactorial inflammatory disease.

## Data Availability

The data presented in this study are available in the article and Appendix A.

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
