# Peer review of "Novel Hydrogen Sulfide Hybrid Derivatives of Keap1-Nrf2 Protein–Protein Interaction Inhibitor Alleviate Inflammation and Oxidative Stress in Acute Experimental Colitis"

_antioxidants, 2023, doi:10.3390/antiox12051062_

Round 1
Reviewer 1 Report
Zhang and co-authors have found that a hybrid derivative (DDO-1901), formed by an inhibitor of Keap1-Nrf2 protein-protein interaction and an H2S-donor moiety, represents a drug candidate more potent than parent drugs, for the UC treatment in both in vitro and in vivo experiments. Indeed, they conclude that the hybridization approach could be an attractive strategy for the treatment of multifactorial inflammatory disease. Overall this is a well-designed and interesting study. The factors at play are sufficiently presented in the introduction. The methodology is clear and uses well-established assays that are adequately explained to a general audience.

Reviewer 2 Report
The work of Zhang et al 2023, ‘Novel Hydrogen Sulfide Hybrid Derivatives of Keap1-Nrf2 2 Protein-protein Interaction Inhibitor Alleviate Inflammation 3 and Oxidative Stress in Acute Experimental Colitis’ is a fascinating paper and offers an interesting insight into the development of novel H2S releasing hybrid molecules for use in the management of colitis. My main comments can be found below.
Please proof-read the document, and ensure the narrative is clearer. Some sentences could be split, for example, ‘Considering the potential beneficial effects of both the Nrf2 pathway and H2S in the treatment of colitis, we began to investigate the possibility that H2S might be used to enhance the efficacy of DDO-1636 and hypothesized that novel hybrids combining both pharmacophore components could display synergistic effects’
This could read, ‘The potential beneficial effects of targeting the Nrf2 pathway and H2S system in the treatment of colitis deserves investigation. Therefore, we have investigated the possibility that H2S could be used to enhance the efficacy of DDO-1636. Our research has led our group to the develop novel H2S releasing hybrids of DDO-1636, that demonstrate improved efficacy in models of colitis’.
Please consider additional refinement of the text and descriptions throughout the manuscript, this needs improving.
Line 233-235. Check the description. You have a stock solution of Na2S of 5mM and use this to generate your standards of 150mM. I assume this is a typo?
Line 453. Please re-write. Something like, Moreover, it was found that DDO-1901 had a concentration dependant impact on Nrf2, NQO1, GCLM, and HO-1 protein expression levels that were more potent than with the parental compounds ADT-OH and DDO-1636 respectively, (Figure 3B).
Reviewer 3 Report
The mentioned study is interesting and brings new possibilities for the treatment of Ulcerative colitis (UC) with hybrid compounds. In in vitro and in vivo experiments, the authors proved that the compound DDO-1901 has a significantly stronger anti-inflammatory effect than the original compounds. The methodological procedures are described in sufficient detail and the results are clearly formulated. However, the English language needs to be corrected significantly.
Major comments:
Why authors decided to apply hybrid drugs to mice by intraperitoneal injection and not by oral route? It could influence absorption and pharmacokinetic of compounds and is not applicable for treatment of patients. Could you explain?
There are significant differences between the ways in which the authors investigated the cytoprotective and anti-inflammatory effect of hybrid compounds in vitro on cells and in vivo on mice, which make it difficult to compare the mechanism of action of the tested compounds.
In in vitro tests on human NCM460 colonocytes, cells were first treated by the compounds and then 3% DSS was added and subsequently viability and other parameters were evaluated. Thus, the preventive effect on the subsequent pro-inflammatory effects of DSS was actually tested. In contrast to this application schedule in the in vivo experiment, the individual compounds were co-administered with 3% DSS for 7 days. See text:
Line 428: The NCM460 cells were pretreated with various compounds (10 μM) for 24 h then exposed to DSS (20 mg/mL) for an additional 12 h. (Fig.2)
Line 325: Mice were challenged with 3% DSS in drinking water for 7 days to induce colitis and treated daily with different drugs by direct intraperitoneal injection.
Line 471: „DSS treatment (20 mg/mL) decreased the activity of SOD and GPx, while pretreatment with DDO-1901 significantly restored the activities of these antioxidant enzymes, enhancing the antioxidant capacity of NCM460 cells (Figure 3C and 3D).„ The conclusions are confusing because DSS was applied after incubation with the DDO-1901 compound. In this treatment scheme DDO-1901 protected cells and prevented the deleterious effects of DSS.
Line 480: „The above study suggested DDO-1901 could alleviate DSS-induced colon injury through activating the antioxidant effect mediated by the Nrf2 signal pathway. „ Re-write sentence, it is not clear.
Line 506 : „Figure 4. (A) Western blot assay for the expression of NOD-like receptor family pyrin domain containing 3 (NLRP3) and caspase-1 p20 in the NCM460 cells after treatment with DDO-1901 and parent drug ADT-OH and DDO-1636. Tubulin was used as an internal reference. (B) Western blot analysis for the expression of NLRP3 and caspase-1 p20 in the NCM460 cells after treatment with various concentrations of DDO-1901 for 24 h.“ Legends to Figure A and B are replaced.
Round 2
Reviewer 3 Report
Authors answered the questions with a good overview of the relevant scientific literature and corrected text reporting unclear conclusions from their experiments. MS has been improved.